# Effects of Community Connectivity on the Spreading Process of Epidemics

**DOI:** 10.3390/e25060849

**Published:** 2023-05-26

**Authors:** Zhongshe Gao, Ziyu Gu, Lixin Yang

**Affiliations:** 1School of Mathematics and Statistics, Tianshui Normal University, Tianshui 741000, China; 2School of Mathematics and Data Science, Shaanxi University of Science & Technology, Xi’an 710021, China

**Keywords:** community structure, epidemic spreading, connection rate

## Abstract

Community structure exists widely in real social networks. To investigate the effect of community structure on the spreading of infectious diseases, this paper proposes a community network model that considers both the connection rate and the number of connected edges. Based on the presented community network, a new SIRS transmission model is constructed via the mean-field theory. Furthermore, the basic reproduction number of the model is calculated via the next-generation matrix method. The results reveal that the connection rate and the number of connected edges of the community nodes play crucial roles in the spreading process of infectious diseases. Specifically, it is demonstrated that the basic reproduction number of the model decreases as the community strength increases. However, the density of infected individuals within the community increases as the community strength increases. For community networks with weak strength, infectious diseases are likely not to be eradicated and eventually will become endemic. Therefore, controlling the frequency and range of intercommunity contact will be an effective initiative to curb outbreaks of infectious diseases throughout the network. Our results can provide a theoretical basis for preventing and controlling the spreading of infectious diseases.

## 1. Introduction

Throughout history, viruses have been a major factor in the development of societies. Describing the mechanisms of epidemics spreading and predicting their epidemiological trends is the long-standing focus of research with the aim of effectively controlling the spread of diseases [1,2,3].

As we know, the study of complex networks has gradually become a hot issue in the field of complexity disciplines. Scholars have made significant contributions to the study of areas such as transportation, social, financial, and biological networks. As research into complex networks has continued, the spreading of computer viruses in computer networks, contagious diseases in social populations, and public opinion and rumors in social networks can have an egregious effect on the development of human society [4,5,6]. Therefore, the behavior of transmission dynamics on complex networks has become one of the research directions of great interest.

Following the introduction of small-world networks and scale-free networks [7,8,9,10], scientists have studied the many networks that exist in the real world and found that complex networks also have characteristics of community structure. Girvan and Newman have introduced community structure properties in many networks and proposed methods for detecting such structures [11]. The community structure means that the nodes in the network are divided into several groups; i.e., the individuals within the groups are relatively tightly connected, and the individuals between the groups are relatively sparsely connected. In real social networks, individuals form groups because of the same characteristics. In turn, scientists study the dynamics of disease transmission in networks by constructing community network models to mirror real networks [12,13,14,15,16,17,18,19,20,21,22,23,24,25].

Ref. [26] defined a parameter to represent the degree of communities to study the effect of community structure on propagation dynamics. Ultimately, it was found that community networks have wider degree distributions and smaller epidemic thresholds compared with random networks. Newman and Girvan defined the modularity coefficient to determine the strength of the community structure network [27]. Later, Salathe and Jones studied the impact of interventions on the spread of disease between communities and found that immunization interventions for individuals within communities in a strong community-structured network were a more effective way to control the spread of disease [28]. Authors investigated two models of community networks with different structures and concluded that the community structure can both prompt and inhibit the spread of viruses, while the intracommunity structure does not affect the spread behavior between communities [29]. Jean-Gabriel studied the emergence of community structures and the network model of structural evolution within communities [30]. Li and Jiang considered a disease model with community heterogeneity and found that the community heterogeneity affects the transmission threshold and disease prevalence rate [31].

Although network propagation models with community structures have been studied very extensively [32,33], most of the models ignore the effect of the number of connected edges within and among communities. To understand the influence of community connectivity on dynamic processes, this paper further studies the epidemic spread model on dynamic networks with community structures. Moreover, demographics play a crucial role in the disease-spreading process. Demographics change the number and internal relationships of individuals in different communities. Therefore, we investigate the effect of the connection rate and the number of connected edges among communities on epidemic propagation in the proposed model.

In fact, in real social networks, the physical contacts among individuals in some communities are closer than in other communities, reflecting the connections among communities. For example, in real social networks, the young-student community interacts with each other more frequently than the elderly-people community. Furthermore, studies have suggested that coupling strength in contact patterns among individuals in different communities has an important effect on epidemic spreading. We firstly construct a new community structure network model with the connection rate and the number of connected edges among communities. Then we establish a novel epidemic spreading model based on this network model and calculate the epidemic threshold.

This paper is organized as follows. In Section 2, a community growth network that considers the number of connected edges is presented. Moreover, we introduce a modularity coefficient to determine the strength of the community structure. Section 3 focuses on the SIR epidemic model via applying the mean-field approach and calculating the basic reproduction number. In Section 4, we give numerical simulations to verify the theoretical results. Finally, Section 5 concludes the paper.

## 2. Community Network Model

The network model generation algorithm with community structure can be constructed as follows [26,29].

Consider a network containing m communities, and each community contains $n_{1},n_{2},\cdots ,n_{m}$ nodes, respectively. Here, we assume that the number of nodes in each community is $n_{i}$.

At the initial moment, the nodes of the network are independent. Each node is connected to $m_{1}$ nodes with probability α of intracommunity links, and every two communities have $m_{2}$ links.

At each time step, each node in the *i*-th community is connected to $m_{3}$ nodes with probability $p_{in}$ in the same community. Moreover, this node is connected to $m_{4}$ nodes with probability $p_{out}$ between different communities. Then, a community network is generated.

In what follows, we discuss the effect of the strength of the community structure on the spreading of the epidemic by introducing a modularity coefficient. The modularity coefficient is defined as [27]
(1)$$Q={\sum_{i}\left[e_{ii}-(\sum_{j}e_{ij}\right)}^{2}],$$
where $e_{ij}$ is the proportion of connected edges in community i and j to the total connected edges in the network.

According to the above network evolution, one can obtain
(2)$$e_{ii}=\frac{n\alpha m_{1}+np_{in}m_{3}}{n\alpha mm_{1}+\frac{1}{2}m(m-1)m_{2}+np_{in}mm_{3}+np_{out}m(m-1)m_{4}}.$$
(3)$$\sum_{j}e_{ij}=e_{ii}+\sum_{j\neq i}e_{ij}=\frac{n\alpha m_{1}+np_{in}m_{3}+(m-1)m_{2}+np_{out}(m-1)m_{4}}{n\alpha mm_{1}+\frac{1}{2}m(m-1)m_{2}+np_{in}mm_{3}+np_{out}m(m-1)m_{4}}.$$

Substituting Equations (2) and (3) into (1), we obtain
(4)$$Q=\frac{(m-1){\left(n\alpha m_{1}+np_{in}m_{3}\right)}^{2}+(n\alpha m_{1}+np_{in}m_{3})A-{\left((m-1)m_{2}+np_{out}(m-1)m_{4}\right)}^{2}}{{\left(n\alpha mm_{1}+\frac{1}{2}m(m-1)m_{2}+np_{in}mm_{3}+np_{out}m(m-1)m_{4}\right)}^{2}}.$$

It can be seen from Equation (4) that the community strength is not only related to the connected rate $p_{in}$, $p_{out}$ but also to the number of intracommunity and intercommunity links. The following numerical simulations are performed. The parameters are taken as m=4, n=500, and α=0.1.

Figure 1 displays the relation between the community strength and the internal and external connection rates of the nodes. From Figure 1, we observe that the same modularity coefficient corresponds to different $p_{in}$ and $p_{out}$. Moreover, it can be seen that the internal connection rate $p_{in}$ increases as the external connection rate $p_{out}$ increases.

Figure 2a shows that when the external connection rate $p_{out}$ is fixed, the modularity coefficient increases as the internal connection rate $p_{in}$ increases. Figure 2b shows that when the internal connection rate $p_{in}$ is fixed, the modularity coefficient decreases as the external connection rate $p_{out}$ increases. It implies that an increase in the frequency of connections between communities leads to a less distinct community structure in the network.

Figure 3a,c display the effect of the number of connected edges of each node within the community on the modularity coefficient. The result is that the higher the number of connected edges, the stronger the community strength. Moreover, Figure 3b,d reflect the increase in the number of connected edges between communities, thus making the community structure inconspicuous. However, Figure 3a,b show that the number of connected edges at the initial moment has a smaller effect on the modularity coefficient. This means that it will not play a decisive role in the strength of the community structure.

## 3. Infectious Disease Dynamics Model and Analysis

### 3.1. Model Description

In this subsection, we consider an epidemic network model with two communities. It is assumed that the individuals in each community will be in three different states: susceptible (S), infected (I), and recovered (R). Parameters $S_{i,k}$, $I_{i,k}$, and $R_{i,k}$ are defined as the density of susceptible, infected, and recovered individuals with degree *k* at time *t*, where i indicates the i-th community, i=1,2. Table 1 denotes the explanation of the key parameters in the transmission of infectious diseases. Here, we assume that all parameters are positive.

In order to further explain, the transformation between the states of the infectious disease in the two communities is described as follows. The Figure 4 shows the infectious disease transmission in complex network with two communities.

Based on the above community network model, the SIR epidemic transmission model is established as the following equations:(5)$$\left\{\begin{array}{l} \frac{dS_{1,k}(t)}{dt}=-\beta_{1}p_{in}kS_{1,k}\Theta^{I_{1}}(t)-\beta_{1}\alpha kS_{1,k}\Theta^{I_{1}}(t)-\gamma_{12}\beta_{2}p_{out}kS_{1,k}\Theta^{I_{2}}(t)+\mu_{1}R_{1,k}(t)\\ \frac{dI_{1,k}(t)}{dt}=\beta_{1}p_{in}kS_{1,k}\Theta^{I_{1}}(t)+\beta_{1}\alpha kS_{1,k}\Theta^{I_{1}}(t)+\gamma_{12}\beta_{2}p_{out}kS_{1,k}\Theta^{I_{2}}(t)-\delta_{1}I_{1,k}(t)\\ \frac{dR_{1,k}(t)}{dt}=\delta_{1}I_{1,k}(t)-\mu_{1}R_{1,k}(t)\\ \frac{dS_{2,k}(t)}{dt}=-\beta_{2}p_{in}kS_{2,k}\Theta^{I_{2}}(t)-\beta_{2}\alpha kS_{2,k}\Theta^{I_{2}}(t)-\gamma_{21}\beta_{1}p_{out}kS_{2,k}\Theta^{I_{1}}(t)+\mu_{2}R_{2,k}(t)\\ \frac{dI_{2,k}(t)}{dt}=\beta_{2}p_{in}kS_{2,k}\Theta^{I_{2}}(t)+\beta_{2}\alpha kS_{2,k}\Theta^{I_{2}}(t)+\gamma_{21}\beta_{1}p_{out}kS_{2,k}\Theta^{I_{1}}(t)-\delta_{2}I_{2,k}(t)\\ \frac{dR_{2,k}(t)}{dt}=\delta_{2}I_{2,k}(t)-\mu_{2}R_{2,k}(t) \end{array}\right.$$
where $n_{i,k}$ represents the total number of individuals of degree k in the i-th community, k=1,2,…,d, d is the maximum degree, $n_{i}$ denotes the whole number of individuals in the i-th community, and N is the total number of individuals in the network, satisfying $N=n_{1}+n_{2}$. We define the power-law degree distribution in the whole network as
(6)$$p(k)=\frac{n_{1,k}+n_{2,k}}{n_{1}+n_{2}}.$$
It is well known that the degree distribution in the i-th community satisfies
(7)$$p_{i}(k)=\frac{n_{i,k}}{n_{i}},$$
and the average degree in the whole network is
(8)$$\begin{array}{l} \langle k\rangle =\sum_{k}kp(k)=\sum_{k}k\cdot \frac{n_{1,k}+n_{2,k}}{n_{1}+n_{2}}=\frac{1}{N}\sum_{k}k(n_{1,k}+n_{2,k})\;.\\ =\frac{1}{N}\sum_{k}k(n_{1}p_{1}(k)+n_{2}p_{2}(k)) \end{array}$$
In Equation (5), $\Theta^{I_{i}}(t)=\frac{1}{\langle k\rangle }\sum_{k}kp(k)\frac{I_{i,k}(t)n_{i,k}}{n_{1,k}+n_{2,k}}$ denotes the probability that a randomly chosen link of a node is connected to an infected node at time t:(9)$$\begin{array}{c} \Theta^{I_{1}}(t)=\frac{1}{\langle k\rangle }\sum_{k}kp(k)\frac{I_{1,k}(t)n_{1,k}}{n_{1,k}+n_{2,k}}=\frac{1}{\langle k\rangle }\sum_{k}k\cdot \frac{n_{1,k}+n_{2,k}}{n_{1}+n_{2}}\cdot \frac{I_{1,k}(t)n_{1,k}}{n_{1,k}+n_{2,k}}=\frac{1}{\langle k\rangle }\sum_{k}k\cdot \frac{I_{1,k}(t)n_{1}p_{1}(k)}{N}\\ =\frac{n_{1}}{N}\frac{1}{\frac{1}{N}({\langle k\rangle }_{1}n_{1}+{\langle k\rangle }_{2}n_{2})}\sum_{k}kp_{1}(k)I_{1,k}(t)=a\sum_{k}kp_{1}(k)I_{1,k}(t) \end{array}$$
where $a=\frac{n_{1}}{{\langle k\rangle }_{1}n_{1}+{\langle k\rangle }_{2}n_{2}}$. Similarly,
(10)$$\Theta^{I_{2}}(t)=b\sum_{k}kp_{2}(k)I_{2,k}(t),$$
where $b=\frac{n_{2}}{{\langle k\rangle }_{1}n_{1}+{\langle k\rangle }_{2}n_{2}}$.

### 3.2. The Basic Reproduction Number

This subsection focuses on the investigation of the basic reproduction number $R_{0}$ via the next-generation matrix scheme. In order to obtain the infection matrix F and the internal evolution matrix V, we translate Equation (5) into the following format:(11)$$\left\{\begin{array}{l} \frac{dI_{1,k}(t)}{dt}=\beta_{1}p_{in}kS_{1,k}\Theta^{I_{1}}(t)+\beta_{1}\alpha kS_{1,k}\Theta^{I_{1}}(t)+\gamma_{12}\beta_{2}p_{out}kS_{1,k}\Theta^{I_{2}}(t)-\delta_{1}I_{1,k}(t)\\ \frac{dI_{2,k}(t)}{dt}=\beta_{2}p_{in}kS_{2,k}\Theta^{I_{2}}(t)+\beta_{2}\alpha kS_{2,k}\Theta^{I_{2}}(t)+\gamma_{21}\beta_{1}p_{out}kS_{2,k}\Theta^{I_{1}}(t)-\delta_{2}I_{2,k}(t)\\ \frac{dS_{1,k}(t)}{dt}=-\beta_{1}p_{in}kS_{1,k}\Theta^{I_{1}}(t)-\beta_{1}\alpha kS_{1,k}\Theta^{I_{1}}(t)-\gamma_{12}\beta_{2}p_{out}kS_{1,k}\Theta^{I_{2}}(t)+\mu_{1}R_{1,k}(t)\\ \frac{dS_{2,k}(t)}{dt}=-\beta_{2}p_{in}kS_{2,k}\Theta^{I_{2}}(t)-\beta_{2}\alpha kS_{2,k}\Theta^{I_{2}}(t)-\gamma_{21}\beta_{1}p_{out}kS_{2,k}\Theta^{I_{1}}(t)+\mu_{2}R_{2,k}(t)\\ \frac{dR_{1,k}(t)}{dt}=\delta_{1}I_{1,k}(t)-\mu_{1}R_{1,k}(t)\\ \frac{dR_{2,k}(t)}{dt}=\delta_{2}I_{2,k}(t)-\mu_{2}R_{2,k}(t) \end{array}\right.$$
where the infection matrix F and the internal evolution matrix V are given:$$\begin{array}{l} F=\left(\begin{array}{c} \beta_{1}p_{in}kS_{1,k}\Theta^{I_{1}}(t)+\beta_{1}\alpha kS_{1,k}\Theta^{I_{1}}(t)+\gamma_{12}\beta_{2}p_{out}kS_{1,k}\Theta^{I_{2}}(t)\\ \beta_{2}p_{in}kS_{2,k}\Theta^{I_{2}}(t)+\beta_{2}\alpha kS_{2,k}\Theta^{I_{2}}(t)+\gamma_{21}\beta_{1}p_{out}kS_{2,k}\Theta^{I_{1}}(t)\\ 0\\ 0\\ 0\\ 0 \end{array}\right)\\ V=\left(\begin{array}{c} \delta_{1}I_{1,k}(t)\\ \delta_{2}I_{2,k}(t)\\ \beta_{1}p_{in}kS_{1,k}\Theta^{I_{1}}(t)+\beta_{1}\alpha kS_{1,k}\Theta^{I_{1}}(t)+\gamma_{12}\beta_{2}p_{out}kS_{1,k}\Theta^{I_{2}}(t)-\mu_{1}R_{1,k}(t)\\ \beta_{2}p_{in}kS_{2,k}\Theta^{I_{2}}(t)+\beta_{2}\alpha kS_{2,k}\Theta^{I_{2}}(t)+\gamma_{21}\beta_{1}p_{out}kS_{2,k}\Theta^{I_{1}}(t)-\mu_{2}R_{2,k}(t)\\ -\delta_{1}I_{1,k}(t)+\mu_{1}R_{1,k}(t)\\ -\delta_{2}I_{2,k}(t)+\mu_{2}R_{2,k}(t) \end{array}\right). \end{array}$$


Then, the infection matrix of the system at the disease-free equilibrium $E_{0}(1,0,0,\ldots ,1,0,0)$ can be obtained:$$\begin{array}{l} F_{11}=\left(\begin{array}{cccc} \beta_{1}(p_{in}+\alpha )\cdot a\cdot p_{1}(1) & \beta_{1}(p_{in}+\alpha )a\cdot 2\cdot p_{1}(2) & \cdots & \beta_{1}(p_{in}+\alpha )\cdot a\cdot d\cdot p_{1}(d)\\ \beta_{1}(p_{in}+\alpha )\cdot 2\cdot a\cdot p_{1}(1) & \beta_{1}(p_{in}+\alpha )\cdot 2\cdot a\cdot 2\cdot p_{1}(2) & \cdots & \beta_{1}(p_{in}+\alpha )\cdot 2\cdot a\cdot d\cdot p_{1}(1)\\ \vdots & \vdots & \ddots & \vdots \\ \beta_{1}(p_{in}+\alpha )\cdot d\cdot a\cdot p_{1}(1) & \beta_{1}(p_{in}+\alpha )\cdot d\cdot a\cdot 2\cdot p_{1}(1) & \cdots & \beta_{1}(p_{in}+\alpha )\cdot d\cdot a\cdot d\cdot p_{1}(d) \end{array}\right)\\ F_{12}=\left(\begin{array}{ccc} \gamma_{12}\beta_{2}p_{out}b\cdot p_{2}(1) & \cdots & \gamma_{12}\beta_{2}p_{out}b\cdot d\cdot p_{2}(d)\\ \vdots & \ddots & \vdots \\ \gamma_{12}\beta_{2}p_{out}d\cdot b\cdot p_{2}(1) & \cdots & \gamma_{12}\beta_{2}p_{out}d\cdot b\cdot d\cdot p_{2}(d) \end{array}\right)\\ F_{21}=\left(\begin{array}{ccc} \gamma_{21}\beta_{1}p_{out}ap_{1}(1) & \cdots & \gamma_{21}\beta_{1}p_{out}a\cdot d\cdot p_{1}(d)\\ \vdots & \ddots & \vdots \\ \gamma_{21}\beta_{1}p_{out}d\cdot a\cdot p_{1}(1) & \cdots & \gamma_{21}\beta_{1}p_{out}d\cdot a\cdot d\cdot p_{1}(d) \end{array}\right)\\ F_{22}=\left(\begin{array}{ccc} \beta_{2}(p_{in}+\alpha )bp_{2}(1) & \cdots & \beta_{2}(p_{in}+\alpha )b\cdot d\cdot p_{2}(d)\\ \vdots & \ddots & \vdots \\ \beta_{2}(p_{in}+\alpha )d\cdot b\cdot p_{2}(1) & \cdots & \beta_{2}(p_{in}+\alpha )d\cdot b\cdot d\cdot p_{2}(d) \end{array}\right) \end{array}$$
where $F_{ij}^{d\times d}$, i=1,2, j=3,…,6, and $F_{ij}^{d\times d}$, i=3,…,6, and j=1,2,…,6 are zero matrix.

The internal evolution matrix of the system at the disease-free equilibrium $E_{0}(1,0,0,\ldots ,1,0,0)$ is given by
$$\begin{array}{ll} V_{11}=\left(\begin{array}{ccc} \delta_{1} & \cdots & 0\\ \vdots & \ddots & \vdots \\ 0 & \cdots & \delta_{1} \end{array}\right) & V_{22}=\left(\begin{array}{ccc} \delta_{2} & \cdots & 0\\ \vdots & \ddots & \vdots \\ 0 & \cdots & \delta_{2} \end{array}\right)\\ V_{31}=\left(\begin{array}{ccc} \beta_{1}(p_{in}+\alpha )b\cdot p_{1}(1) & \cdots & \beta_{1}(p_{in}+\alpha )b\cdot d\cdot p_{1}(d)\\ \vdots & \ddots & \vdots \\ \beta_{1}(p_{in}+\alpha )d\cdot b\cdot p_{1}(1) & \cdots & \beta_{1}(p_{in}+\alpha )d\cdot b\cdot d\cdot p_{1}(d) \end{array}\right) & \\ V_{32}=\left(\begin{array}{ccc} \gamma_{12}\beta_{2}p_{out}b\cdot p_{2}(1) & \cdots & \gamma_{12}\beta_{2}p_{out}b\cdot d\cdot p_{2}(d)\\ \vdots & \ddots & \vdots \\ \gamma_{12}\beta_{2}p_{out}d\cdot b\cdot p_{2}(1) & \cdots & \gamma_{12}\beta_{2}p_{out}d\cdot b\cdot d\cdot p_{2}(d) \end{array}\right) & \\ V_{35}=\left(\begin{array}{ccc} -\mu_{1} & \cdots & 0\\ \vdots & \ddots & \vdots \\ 0 & \cdots & -\mu_{1} \end{array}\right) & \\ V_{41}=\left(\begin{array}{ccc} \gamma_{21}\beta_{1}p_{out}ap_{1}(1) & \cdots & \gamma_{21}\beta_{1}p_{out}a\cdot d\cdot p_{1}(d)\\ \vdots & \ddots & \vdots \\ \gamma_{21}\beta_{1}p_{out}d\cdot a\cdot p_{1}(1) & \cdots & \gamma_{21}\beta_{1}p_{out}d\cdot a\cdot d\cdot p_{1}(d) \end{array}\right) & \\ V_{42}=\left(\begin{array}{ccc} \beta_{2}(p_{in}+\alpha )bp_{2}(1) & \cdots & \beta_{2}(p_{in}+\alpha )b\cdot d\cdot p_{2}(d)\\ \vdots & \ddots & \vdots \\ \beta_{2}(p_{in}+\alpha )d\cdot b\cdot p_{2}(1) & \cdots & \beta_{2}(p_{in}+\alpha )d\cdot b\cdot d\cdot p_{2}(d) \end{array}\right) & \\ V_{46}=\left(\begin{array}{ccc} -\mu_{2} & \cdots & 0\\ \vdots & \ddots & \vdots \\ 0 & \cdots & -\mu_{2} \end{array}\right)\;V_{51}=\left(\begin{array}{ccc} -\delta_{1} & \cdots & 0\\ \vdots & \ddots & \vdots \\ 0 & \cdots & -\delta_{1} \end{array}\right)\;V_{55}=\left(\begin{array}{ccc} \mu_{1} & \cdots & 0\\ \vdots & \ddots & \vdots \\ 0 & \cdots & \mu_{1} \end{array}\right) & \\ V_{62}=\left(\begin{array}{ccc} -\delta_{2} & \cdots & 0\\ \vdots & \ddots & \vdots \\ 0 & \cdots & -\delta_{2} \end{array}\right)\;V_{66}=\left(\begin{array}{ccc} \mu_{2} & \cdots & 0\\ \vdots & \ddots & \vdots \\ 0 & \cdots & \mu_{2} \end{array}\right) & \end{array}$$
where $V_{1j}^{d\times d}$, j=2,…,6, $V_{2j}^{d\times d}$, j=1,3,…,6, $V_{3j}^{d\times d}$,j=3,4,6, $V_{4j}^{d\times d}$, j=3,4,5, $V_{5j}^{d\times d}$, j=2,3,4,6, $V_{6j}^{d\times d}$, and j=1,3,4,5 are zero matrices according to the principle of the next-generation matrix, where $R_{0}=\rho (FV^{-1})$, which denotes the spectral radius of the matrix $FV^{-1}$.

Defining $C=FV^{-1}$, we have $C_{1j}^{d\times d}$, (j=3,…,6), $C_{2j}^{d\times d}$, (j=3,…,6), $C_{ij}^{d\times d}$, and (i=3,…,6,j=1,…,6) are zero matrices,

$C_{11}^{d\times d}=\frac{\beta_{1}(p_{in}+\alpha )a}{\delta_{1}}T_{1}^{d\times d}$, $C_{12}^{d\times d}=\frac{\gamma_{12}\beta_{2}p_{out}b}{\delta_{2}}T_{2}^{d\times d}$, $C_{21}^{d\times d}=\frac{\gamma_{21}\beta_{1}p_{out}a}{\delta_{1}}T_{1}^{d\times d}$,$C_{22}^{d\times d}=\frac{\beta_{2}(p_{in}+\alpha )b}{\delta_{2}}T_{2}^{d\times d}$

where $T_{i}^{d\times d}=\left(\begin{array}{ccc} p_{i}(1) & \cdots & dp_{i}(d)\\ \vdots & \ddots & \vdots \\ dp_{i}(1) & \cdots & d^{2}p_{i}(d) \end{array}\right)$.

In the following, we determine the matrix C of the spectral radius:$$C=\left(\begin{array}{cc} C_{11}^{d\times d} & C_{12}^{d\times d}\\ C_{21}^{d\times d} & C_{22}^{d\times d} \end{array}\right).$$

It is found that the matrix C has 2d−2 eigenvalues equal to 0 by applying similar transformation to the matrix C. The remaining two eigenvalues satisfy
(12)$$\left|\begin{array}{cc} \frac{\beta_{1}(p_{in}+\alpha )a{\langle k^{2}\rangle }_{1}}{\delta_{1}}-x & \frac{\gamma_{12}\beta_{2}p_{out}b{\langle k^{2}\rangle }_{2}}{\delta_{2}}\\ \frac{\gamma_{21}\beta_{1}p_{out}a{\langle k^{2}\rangle }_{1}}{\delta_{1}} & \frac{\beta_{2}(p_{in}+\alpha )b{\langle k^{2}\rangle }_{2}}{\delta_{2}}-x \end{array}\right|=0.$$

From the determinant (12), the characteristic equation can be obtained as
(13)$$\left(\frac{\beta_{1}(p_{in}+\alpha )a{\langle k^{2}\rangle }_{1}}{\delta_{1}}-x\right)\left(\frac{\beta_{2}(p_{in}+\alpha )b{\langle k^{2}\rangle }_{2}}{\delta_{2}}-x\right)-\frac{\gamma_{12}\beta_{2}p_{out}b{\langle k^{2}\rangle }_{2}}{\delta_{2}}\cdot \frac{\gamma_{21}\beta_{1}p_{out}a{\langle k^{2}\rangle }_{1}}{\delta_{1}}=0.$$

Therefore, the discriminant of the roots of quadratic Equation (13) is
(14)$$\begin{array}{ll} \Delta & ={\left(\frac{\beta_{1}(p_{in}+\alpha )a{\langle k^{2}\rangle }_{1}}{\delta_{1}}+\frac{\beta_{2}(p_{in}+\alpha )b{\langle k^{2}\rangle }_{2}}{\delta_{2}}\right)}^{2}\\ & \;-4\left(\frac{\beta_{1}(p_{in}+\alpha )a{\langle k^{2}\rangle }_{1}}{\delta_{1}}\cdot \frac{\beta_{2}(p_{in}+\alpha )b{\langle k^{2}\rangle }_{2}}{\delta_{2}}-\frac{\gamma_{12}\beta_{2}p_{out}b{\langle k^{2}\rangle }_{2}}{\delta_{2}}\cdot \frac{\gamma_{21}\beta_{1}p_{out}a{\langle k^{2}\rangle }_{1}}{\delta_{1}}\right)\;.\\ & ={\left(\frac{\beta_{1}(p_{in}+\alpha )a{\langle k^{2}\rangle }_{1}}{\delta_{1}}-\frac{\beta_{2}(p_{in}+\alpha )b{\langle k^{2}\rangle }_{2}}{\delta_{2}}\right)}^{2}+4\frac{\gamma_{12}\beta_{2}p_{out}b{\langle k^{2}\rangle }_{2}}{\delta_{2}}\cdot \frac{\gamma_{21}\beta_{1}p_{out}a{\langle k^{2}\rangle }_{1}}{\delta_{1}}>0 \end{array}$$

The roots of the quadratic equation; that is, the eigenvalues of the matrix C, are
(15)$$x_{1,2}=\frac{\left(\frac{\beta_{1}(p_{in}+\alpha )a{\langle k^{2}\rangle }_{1}}{\delta_{1}}+\frac{\beta_{2}(p_{in}+\alpha )b{\langle k^{2}\rangle }_{2}}{\delta_{2}}\right)\pm \sqrt{\Delta }}{2}.$$

Thus, the basic reproduction number of the whole network is expressed as [34]
(16)$$\begin{array}{ll} R_{0}= & \frac{1}{2}\left(\frac{\beta_{1}(p_{in}+\alpha )a{\langle k^{2}\rangle }_{1}}{\delta_{1}}+\frac{\beta_{2}(p_{in}+\alpha )b{\langle k^{2}\rangle }_{2}}{\delta_{2}}\right)\\ & +\frac{1}{2}\sqrt{{\left(\frac{\beta_{1}(p_{in}+\alpha )a{\langle k^{2}\rangle }_{1}}{\delta_{1}}-\frac{\beta_{2}(p_{in}+\alpha )b{\langle k^{2}\rangle }_{2}}{\delta_{2}}\right)}^{2}+4\frac{\gamma_{12}\beta_{2}p_{out}b{\langle k^{2}\rangle }_{2}}{\delta_{2}}\cdot \frac{\gamma_{21}\beta_{1}p_{out}a{\langle k^{2}\rangle }_{1}}{\delta_{1}}}. \end{array}$$

Equation (16) shows that the probability of being connected in the inner and outer communities plays a positive role in the basic reproduction number. According to the above analysis, the following theorem is obtained.

**Theorem** **1:***For system (5), if the basic reproduction number*$\;R_{0}<1$*, the disease-free equilibrium is stable and the disease will die out in two communities. If the basic reproduction number *$R_{0}>1$, *the disease-free equilibrium is unstable and the epidemic can occur in two communities.*

## 4. The Impact of Community Structure on the Spreading of Infectious Diseases

In the following, we study the influence of community structure on the process of spreading infectious diseases. Here, we focus on the variation in the basic reproduction number and infection density.

### 4.1. The Influence of the Connection Rate on the Basic Reproduction Number

Firstly, we study the impact of the connection rate within and between the communities on the basic reproduction number. The parameters are taken as $\langle k_{1}\rangle =9.08$, $\langle k_{2}\rangle =8.83$, $\delta_{1}=0.05$, $\delta_{2}=0.1$, $\gamma_{12}=0.5$, and $\gamma_{21}=0.35$.

Fix parameter $p_{out}=0.2$; Figure 5 shows the spatial-temporal distribution of the basic reproduction number in the network with the infection and connection rates within the community. Let parameter $p_{in}=0.2$; Figure 6 shows the spatial-temporal distribution of the basic reproduction number in the network with the infection rate within the community and the external connection rate. These figures represent that the internal connection rate and the external connection rate are positively correlated with the basic reproduction number. Comparing Figure 5 and Figure 6, it is observed that the external connection rate has a greater impact on the basic reproduction number for the network.

### 4.2. The Influence of the Connection Rate and Connected Edges on Infection Density

In this section, we discuss the impact of the connection rate and the number of connected edges within and outside the communities on the transmission of infectious diseases. The parameter takes the value n=100, and the initial infected node I(0)=1.

Figure 7 shows the variation in infection density with the internal connection rate in the community when $p_{out}=0.2$. One can see that the greater the internal connection rate in the community, the more rapidly the density of infected individuals grows. The reason is that individual-to-individual contact becomes frequent within the community, and if an infected individual appears at this time, it is more likely for the infected individual to spread the virus to susceptible individuals, which leads to an increase in the size of the infection in the community as a whole. It is also found that $p_{in}$ can affect the peak time in single communities.

Figure 8 describes the variation in the infection density with the external connection rate when $p_{in}=0.5$. It can be observed that if the external connection rate $p_{out}$ increases, it means that intercommunity contact increases, which then also increases the rate and scale of the spreading of infectious diseases throughout the network. In reality, the multiple cross-provincial movements of individuals lead to frequent external connections, with the result that infectious diseases spread on a large scale across the country. Thus, it seems that reducing the number of trips is an effective way to control the spreading of infectious diseases.

Observing the red solid line and the blue dashed part in Figure 7 and Figure 8, it is known that when $p_{out}$ increases by 0.05, the density of infection can increase by about 0.03. When $p_{in}$ increases by 0.15, the density of infection only increases by less than 0.02. Based on the above data, one obtains that intercommunity connection plays a crucial role in the spread of infectious diseases.

In what follows, we investigate the effect of the number of connected edges within communities and between communities on the transmission of infectious diseases, respectively.

Figure 9 shows the variation in infection density with the number of connected edges within the community for the case of $R_{0}<1$. It can be observed when the number of new connected edges $m_{3}$ of nodes within the community increases, the speed of infection and the size of the outbreak increase. Thus, it can be concluded that the more susceptible people an infected person in the community comes in contact with, the wider the spread of the infection within the network, and ultimately the greater the number of infected people.

Figure 10 indicates the variation in infection density with the number of connected edges between communities for the case of $R_{0}>1$. From this figure, one can observe that the higher the number of connected edges between communities, the faster the speed of the infection. When infected individuals move between communities, they come into contact with many individuals from various other communities. The result shows that the speed of the spreading of infected individuals, as well as the final size of the infectious disease, also increases.

The above analysis implies that reducing the movement of individuals across provinces can control large-scale outbreaks of infectious diseases in certain conditions.

In addition, we consider the number of nodes within the communities is $n_{1}=n_{2}=100$. The number of new connected edges added by individuals at each time step within the two communities is $m_{3}^{1}=8$, $m_{3}^{2}=5$. Figure 11 gives the curve of the density of each state of the infectious disease with time for different community structures.

Comparing the infection density curves in Figure 11, the basic reproduction number $R_{0}>1$, the infection density of the network with more connected nodes within the community, will continue to increase until it reaches a steady state. This implies that the virus is not spreading massively, so the curve of the infection density in the graph gradually decreases to a steady state.

As we know, intercommunity activities play an important role in the spreading of infectious diseases. Here we combine Equations (4) and (16) to investigate the effect of community strength on the spread of infectious diseases. Figure 12 shows the variation curve of community strength with the basic reproduction number, where α=0.1 and $p_{in}=0.2$.

From Figure 12, we can find that the basic reproduction number decreases as the modularity coefficient Q increases. This implies that the stronger the community structure, the less widespread the spread of infectious diseases will be. In fact, this conclusion holds, but only if the internal connection rate *p-in* and the number of inner edges are constant and the external connection rate *p-out* and the number of intracommunity edges are decreasing. Therefore, the possibility of a virus outbreak at the social level as a whole is small. When the community structure is weak enough, the basic regeneration number is greater than 1, and eventually the epidemic evolves into an endemic disease.

From the above analysis, we can summarize that the strength of the community can affect the final form of the infectious disease, i.e., whether it is eradicated or forms an endemic disease.

## 5. Conclusions

This paper presents a modified community network model considering the connection rate of nodes and the number of connected edges between communities. Based on this model, an SIR infectious disease transmission model with two community structures is constructed, which is closer to real networks. Furthermore, the transmission dynamics of epidemics are analyzed via theoretical analysis and numerical simulation. Our results show that the higher the frequency of susceptible nodes being infected due to the frequent contact of individuals within the community, the easier the infectious disease will spread within the community. When the external connection rate of the community increases, the movement of people between communities becomes frequent, which leads to the transmission of the infection throughout the network. In addition, when the external connected edges of communities increase, the infectious disease spreads rapidly between communities, and the number of infected individuals in the entire network increases massively.

In conclusion, controlling unnecessary movement of people and minimizing visits to crowded spaces are effective measures to curb the transmission of infectious diseases. This is also in line with our current epidemic prevention policy of not going out of town unless necessary and reducing the number of trips. In the future, we intend to investigate mathematical properties of the epidemic model and further explore the impact of data-driven models for multiplex networks.

## Figures and Tables

**Figure 1 entropy-25-00849-f001:**
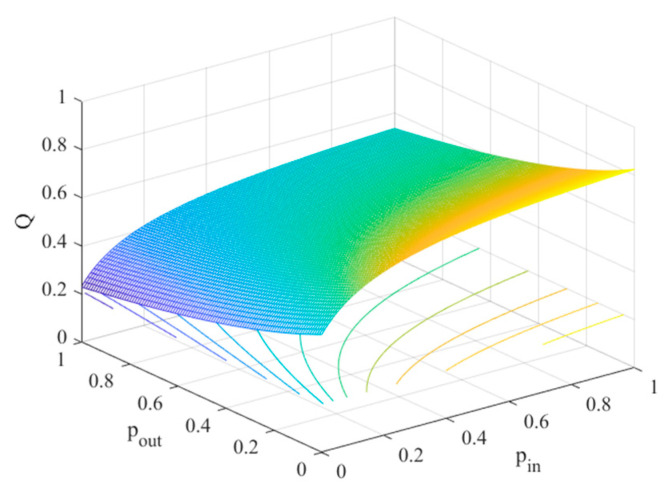
The relationships between Q and $p_{in}$, $p_{out}$.

**Figure 2 entropy-25-00849-f002:**
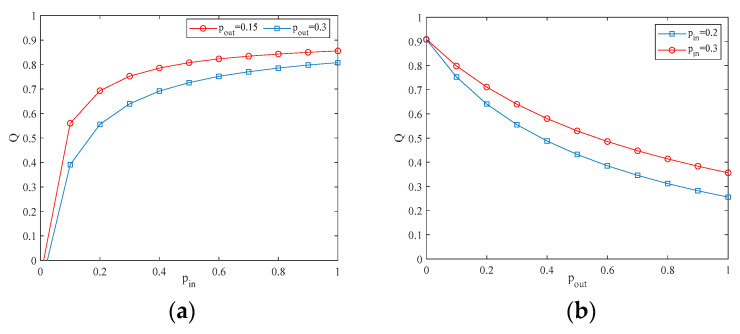
(**a**,**b**) The relationship between Q and the internal connection rates.

**Figure 3 entropy-25-00849-f003:**
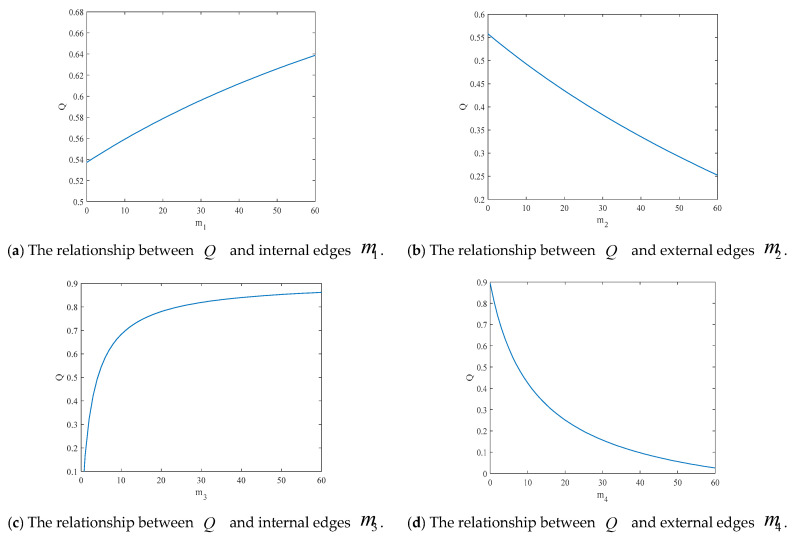
The relationship between the modularity coefficient Q and the connected edges.

**Figure 4 entropy-25-00849-f004:**
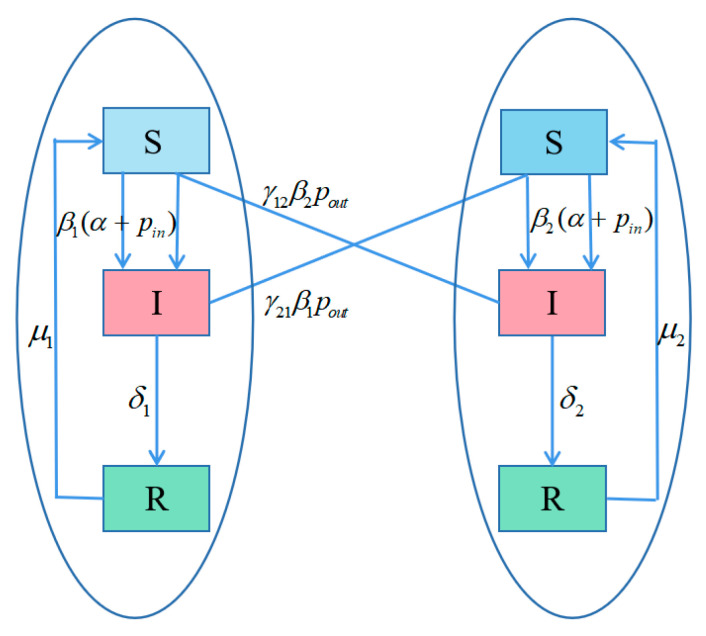
The flow chart of the infectious disease transmission in two community structure networks.

**Figure 5 entropy-25-00849-f005:**
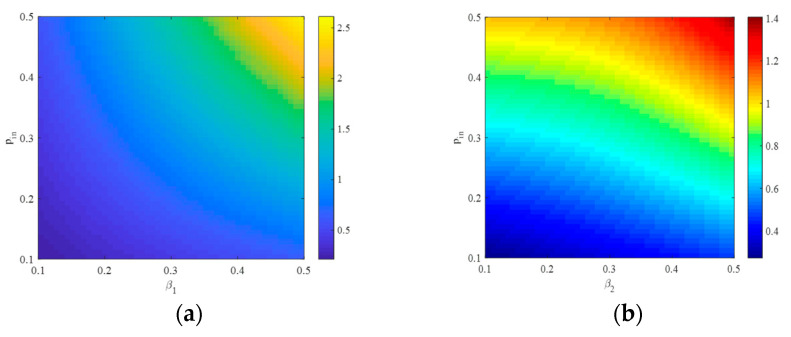
(**a**,**b**) The spatial-temporal distribution of $R_{0}$ with β and $p_{in}$.

**Figure 6 entropy-25-00849-f006:**
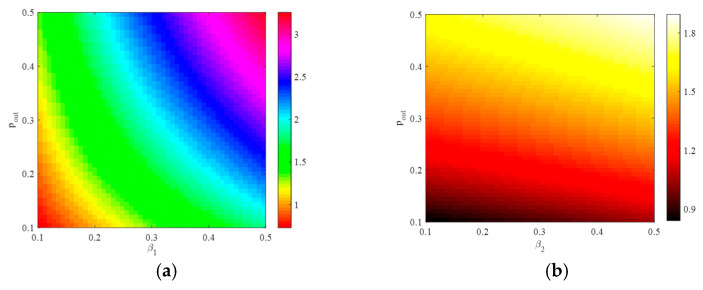
(**a**,**b**) The spatial-temporal distribution of $R_{0}$ with β and $p_{out}$.

**Figure 7 entropy-25-00849-f007:**
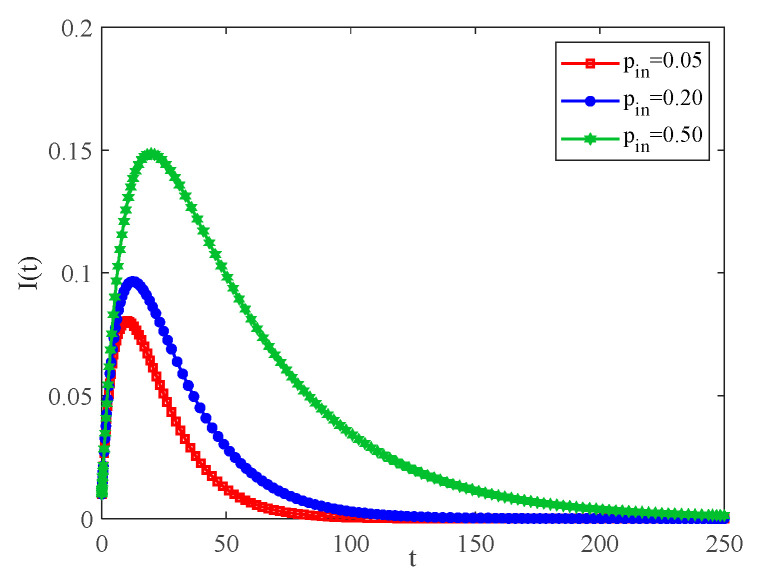
The variation curves of infection density with time for different $p_{in}$, $p_{out}=0.2$.

**Figure 8 entropy-25-00849-f008:**
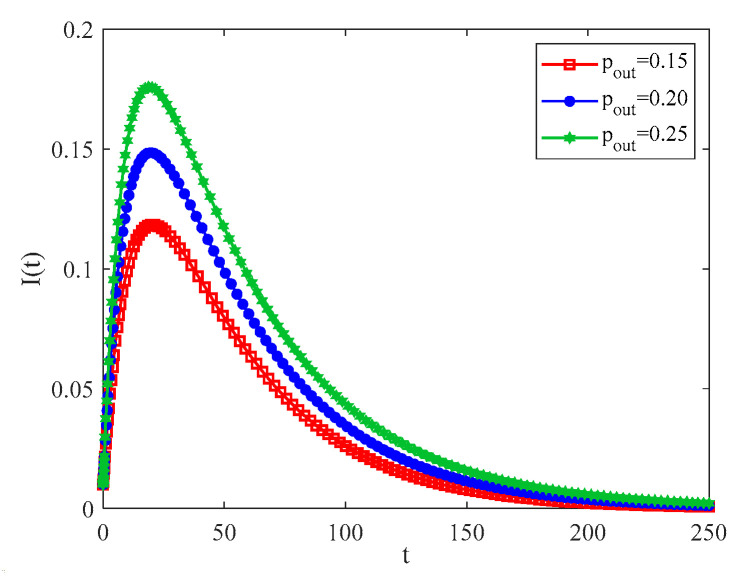
The evolutions of infection density with time for different $p_{out},p_{in}=0.5$.

**Figure 9 entropy-25-00849-f009:**
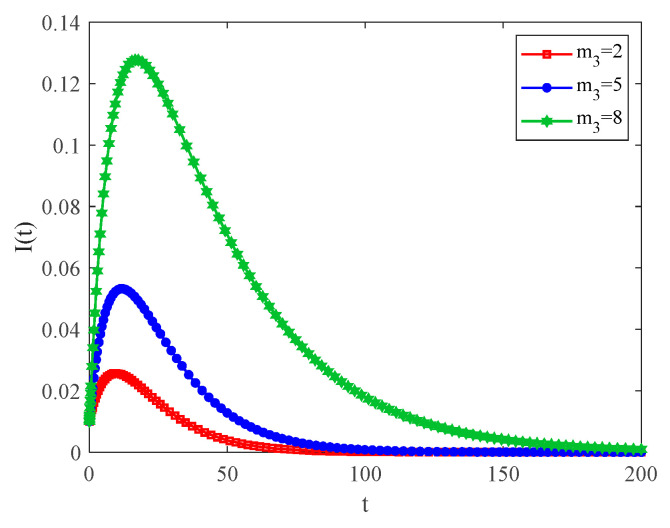
The variation curves of infection density with time for different $m_{3}$ and $R_{0}<1$.

**Figure 10 entropy-25-00849-f010:**
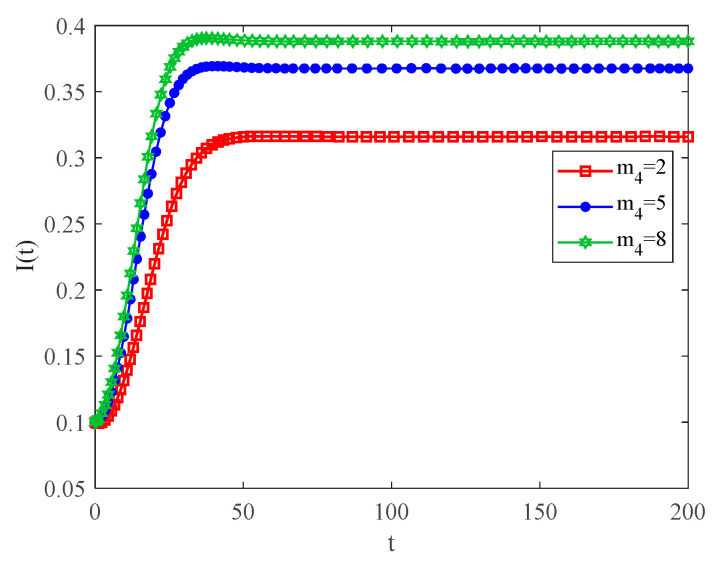
The variation curves of infection density with time for different $m_{4}$ and $R_{0}>1$.

**Figure 11 entropy-25-00849-f011:**
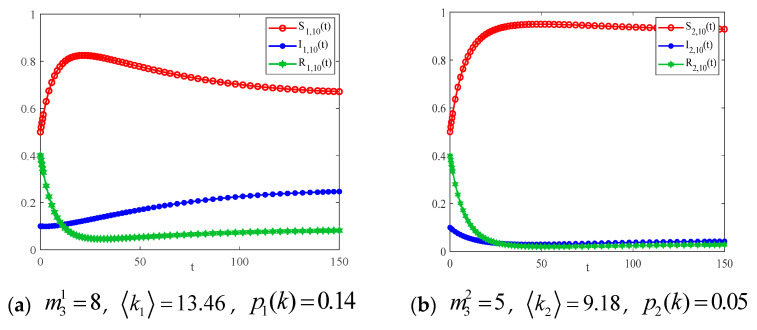
The variation in S(t), I(t), and R(t) with time.

**Figure 12 entropy-25-00849-f012:**
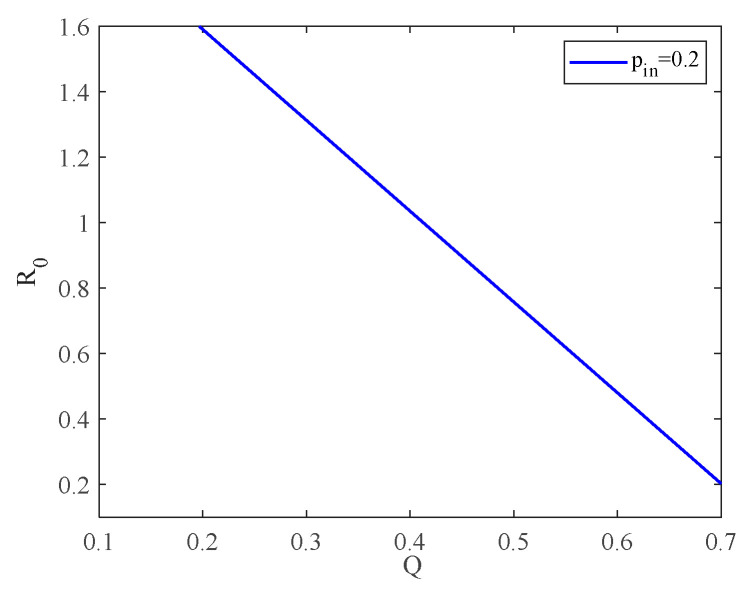
The variation in $R_{0}$ with Q.

**Table 1 entropy-25-00849-t001:** The explanation of all the system parameters.

Parameter	Explanation
$\beta_{i}$	Infection rate in the i-th community
α	Probability of nodes being connected in the inner community at the initial moment
$p_{in}$	Probability of nodes being connected in the inner community at each time step
$p_{out}$	Probability of nodes being connected in the outer community at each time step
$\delta_{i}$	Recovery rate of infected in the i-th community
$\mu_{i}$	Probability of recovered reverting to susceptible
$\gamma_{ij}$	Coefficient factor affecting the infection rate in the outer community

## Data Availability

The datasets generated and analyzed during the current study are available from the corresponding author on reasonable request.

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
