# Peer review of "Effects of Community Connectivity on the Spreading Process of Epidemics"

_entropy, 2023, doi:10.3390/e25060849_

Round 1

Reviewer 1 Report

In this paper the authors proposed a community network model considering both the connection rate and the number of connected edges, and based on the community network they investigate the effect of the community strength on epidemic spreading process. I think the topic is interesting and the paper is well organized. However, there are some shortcomings listed as follows.

 1. In Section 1, I suggest the authors further underline the motivations and the findings.

 2. In Section 1, at the end of 2th sentence of 2th paragraph, it is better to give several related literatures citation which topics involved in the spreading of computer viruses, contagious diseases, and public opinion etc. Moreover, I suggest to delete the last sentence in 4th paragraph to make the context clear.

 3. In Section 2, I suggest to rewrite 4th paragraph “At the time step,…” since it is not clear and difficult to understand. Moreover, here please give the end condition of the community construction.

 4. The author should explain the meaning of the modularity coefficient Q and the terms e_ij and e_ii in Eq. (1).

 5. Please check Eq. (2) and Eq. (3), maybe the variable n should be n_i, moreover the variable n seems to be not defined.

 6. On page 7, check the sentence “F_ij^dxd, i=1,…,6, j=3,…,6 are zero matrix”. The values of the subscripts i,j seem to be incorrect.

 7. After working out R_0, Why not further discuss the asymptotic stability of the system? Give some explanations.

 8.  Fig.7-Fig.11 are not clear, especially the X-axis and the Y-axis, I suggest to increase the resolution and the font size.

Minor editing of English language required.

Author Response

Responses to the Reviewer #1’s comments:

In this paper the authors proposed a community network model considering both the connection rate and the number of connected edges, and based on the community network they investigate the effect of the community strength on epidemic spreading process. I think the topic is interesting and the paper is well organized. However, there are some shortcomings listed as follows. 

(1) In Section 1, I suggest the authors further underline the motivations and the findings.

Response: Thank you for your kind comments. We have added the motivations and the findings. in the revised version.

(2) In Section 1, at the end of 2th sentence of 2th paragraph, it is better to give several related literatures citation which topics involved in the spreading of computer viruses, contagious diseases, and public opinion etc. Moreover, I suggest delete the last sentence in 4th paragraph to make the context clear.

Response: Following the suggestion of the reviewer, we have added the according references in the revised version.

(3)In Section 2, I suggest to rewrite 4th paragraph “At the time step,…” since it is not clear and difficult to understand. Moreover, here please give the end condition of the community construction.

Response: According to the reviewer’s comment, we have rewritten the expressions of the 4th paragraph in the revised paper. Furthermore, if we fix the total number of nodes and communities, and other parameters, one can construct a community model. That is, the end condition of the community construction changes with the values of parameters.

 (4) The author should explain the meaning of the modularity coefficient Q and the terms e_ij and e_ii in Eq. (1).

Response: Following the suggestion of the reviewer, we have added the meaning of the modularity coefficient. In addition, the modularity coefficient Q is explained in the following reference in detail.

  • Newman M, Girvan M. Finding and evaluating community structure in networks[J]. Physical Review E, 2004, 69(2): 026113.

In general, the modularity coefficient is defined as

where denotes the proportion of the connecting edges between the community and the total connecting edges in the network.

Based on the evolution of complex network, we can obtain the description of

The value range of modularity coefficient Q is , when it approaches 0, the network is a random network; when it approaches 1, the network has a strong community structure. In this paper, the community structure is denoted by the ration of external links and internal links. Thus, a smaller ratio corresponds to sparse external links thus a stronger community structure.

(5) Please check Eq. (2) and Eq. (3), maybe the variable n should be n_i, moreover the variable n seems to be not defined.

Response: Following the suggestion of the reviewer, we find that Eq.(2) and (3) are correct. In fact, the variable n can be rewritten ni. Without loss of generality, this paper assumes that the number of nodes in each community is the same. Thus, the variable n is defined as the number of nodes in every community.

(6) On page 7, check the sentence “F_ij^dxd, i=1,…,6, j=3,…,6 are zero matrix”. The values of the subscripts i,j seem to be incorrect.

Response: We agree with the reviewer’s comments, the values of the subscripts are really incorrect. We have revised the description.

(7) After working out R_0, Why not further discuss the asymptotic stability of the system? Give some explanations.

Response: It is really true as reviewer commented that it is necessary to further discuss the asymptotic stability of the system. In fact, the approach of analyzing asymptotic stability of the model is similar with many existing studies. So we omit here. But we would like to stress that this paper focuses on the effect of community connectivity on the spreading process of epidemics.

For example, the stability of the system can be described as follows:

Theorem 1: If, the disease-free equilibriumof the epidemic model is globally stable

proof:suppose is the left eigenvector of matrix ,one can construct the Lypaunov function

Denote  is the identity matrix,when , then ,one can obtain

Thus,

. That is, is the unique invariant subset of the set. Based on the LaSalle invariance principle, the disease-free equilibrium is globally asymptotically stable.

(8) Fig.7-Fig.11 are not clear, especially the X-axis and the Y-axis, I suggest to increase the resolution and the font size.

Response: According to the reviewer’s comments, we have revised Fig.7-Fig.11 in the revised version.

Responses to the Reviewer #1’s comments:

In this paper the authors proposed a community network model considering both the connection rate and the number of connected edges, and based on the community network they investigate the effect of the community strength on epidemic spreading process. I think the topic is interesting and the paper is well organized. However, there are some shortcomings listed as follows. 

(1) In Section 1, I suggest the authors further underline the motivations and the findings.

Response: Thank you for your kind comments. We have added the motivations and the findings. in the revised version.

(2) In Section 1, at the end of 2th sentence of 2th paragraph, it is better to give several related literatures citation which topics involved in the spreading of computer viruses, contagious diseases, and public opinion etc. Moreover, I suggest delete the last sentence in 4th paragraph to make the context clear.

Response: Following the suggestion of the reviewer, we have added the according references in the revised version.

(3)In Section 2, I suggest to rewrite 4th paragraph “At the time step,…” since it is not clear and difficult to understand. Moreover, here please give the end condition of the community construction.

Response: According to the reviewer’s comment, we have rewritten the expressions of the 4th paragraph in the revised paper. Furthermore, if we fix the total number of nodes and communities, and other parameters, one can construct a community model. That is, the end condition of the community construction changes with the values of parameters.

 (4) The author should explain the meaning of the modularity coefficient Q and the terms e_ij and e_ii in Eq. (1).

Response: Following the suggestion of the reviewer, we have added the meaning of the modularity coefficient. In addition, the modularity coefficient Q is explained in the following reference in detail.

  • Newman M, Girvan M. Finding and evaluating community structure in networks[J]. Physical Review E, 2004, 69(2): 026113.

In general, the modularity coefficient is defined as

where denotes the proportion of the connecting edges between the community and the total connecting edges in the network.

Based on the evolution of complex network, we can obtain the description of

The value range of modularity coefficient Q is , when it approaches 0, the network is a random network; when it approaches 1, the network has a strong community structure. In this paper, the community structure is denoted by the ration of external links and internal links. Thus, a smaller ratio corresponds to sparse external links thus a stronger community structure.

(5) Please check Eq. (2) and Eq. (3), maybe the variable n should be n_i, moreover the variable n seems to be not defined.

Response: Following the suggestion of the reviewer, we find that Eq.(2) and (3) are correct. In fact, the variable n can be rewritten ni. Without loss of generality, this paper assumes that the number of nodes in each community is the same. Thus, the variable n is defined as the number of nodes in every community.

(6) On page 7, check the sentence “F_ij^dxd, i=1,…,6, j=3,…,6 are zero matrix”. The values of the subscripts i,j seem to be incorrect.

Response: We agree with the reviewer’s comments, the values of the subscripts are really incorrect. We have revised the description.

(7) After working out R_0, Why not further discuss the asymptotic stability of the system? Give some explanations.

Response: It is really true as reviewer commented that it is necessary to further discuss the asymptotic stability of the system. In fact, the approach of analyzing asymptotic stability of the model is similar with many existing studies. So we omit here. But we would like to stress that this paper focuses on the effect of community connectivity on the spreading process of epidemics.

For example, the stability of the system can be described as follows:

Theorem 1: If, the disease-free equilibriumof the epidemic model is globally stable

proof:suppose is the left eigenvector of matrix ,one can construct the Lypaunov function

Denote  is the identity matrix,when , then ,one can obtain

Thus,

. That is, is the unique invariant subset of the set. Based on the LaSalle invariance principle, the disease-free equilibrium is globally asymptotically stable.

(8) Fig.7-Fig.11 are not clear, especially the X-axis and the Y-axis, I suggest to increase the resolution and the font size.

Response: According to the reviewer’s comments, we have revised Fig.7-Fig.11 in the revised version.

Reviewer 2 Report

The authors construct here a community network model (a network with a particular community structure) and investigate the effects of a SIR-type infectious disease spreading process taking place on top of such a network. A case with two communities is analysed in detail. The outputs are reasonable, i.e., in agreement in connection with what one could expect and the paper is clearly presented.

Only a minor observation. I think that the two sentences 

“Although network propagation models with community structure have been studied very extensively [33]. However, most of the models ignore the effect of the number of connected edges within and among communities.”

starting on line 22 on page 2 should be slightly modified to

“Although network propagation models with community structure have been studied very extensively [33], most of the models ignore the effect of the number of connected edges within and among communities.”

Indeed, the verb is missing in the first sentence; this was probably due to a misprint (a change occurred while writing).

Author Response

The authors construct here a community network model (a network with a particular community structure) and investigate the effects of a SIR-type infectious disease spreading process taking place on top of such a network. A case with two communities is analyzed in detail. The outputs are reasonable, i.e., in agreement in connection with what one could expect and the paper is clearly presented.

Only a minor observation. I think that the two sentences 

(1) “Although network propagation models with community structure have been studied very extensively [33]. However, most of the models ignore the effect of the number of connected edges within and among communities.”

starting on line 22 on page 2 should be slightly modified to

“Although network propagation models with community structure have been studied very extensively [33], most of the models ignore the effect of the number of connected edges within and among communities.”

Indeed, the verb is missing in the first sentence; this was probably due to a misprint (a change occurred while writing).

Response: Thanks the reviewer for pointing our mistake. We have revised the inaccurate sentences in the revised version.
